# iAgent: LLM Agent as a Shield between User and Recommender Systems

## Abstract

Traditional recommender systems usually take the user-platform paradigm, where users are directly exposed under the control of the platform's recommendation algorithms. However, the defect of recommendation algorithms may put users in very vulnerable positions under this paradigm. First, many sophisticated models are often designed with commercial objectives in mind, focusing on the platform's benefits, which may hinder their ability to protect and capture users' true interests. Second, these models are typically optimized using data from all users, which may overlook individual user's preferences. Due to these shortcomings, users may experience several disadvantages under the traditional user-platform direct exposure paradigm, such as lack of control over the recommender system, potential manipulation by the platform, echo chamber effects, or lack of personalization for less active users due to the dominance of active users during collaborative learning. Therefore, there is an urgent need to develop a new paradigm to protect user interests and alleviate these issues. Recently, some researchers have introduced LLM agents to simulate user behaviors, these approaches primarily aim to optimize platform-side performance, leaving core issues in recommender systems unresolved. To address these limitations, we propose a new user-agent-platform paradigm, where agent serves as the protective shield between user and recommender system that enables indirect exposure. To this end, we first construct four recommendation datasets, denoted as INSTRUCTREC, along with user instructions for each record. To understand user's intention, we design an Instruction-aware Agent (iAgent) capable of using tools to acquire knowledge from external environments. Moreover, we introduce an Individual Instruction-aware Agent ($i^2$Agent), which incorporates a dynamic memory mechanism to optimize from individual feedback. Results on four INSTRUCTREC datasets demonstrate that $i^2$Agent consistently achieves an average improvement of 16.6% over SOTA baselines across ranking metrics. Moreover, $i^2$Agent mitigates echo chamber effects and effectively alleviates the model bias in disadvantaged users (less-active), serving as a shield between user and recommender systems. Datasets and code are publicly available at the URL[1].

## 1 Introduction

Over the past decades, recommender systems have been extensively applied across various platforms to provide personalized services to users. In the traditional ecosystem of recommender systems, the recommendation models are predominantly delivered through a user-platform paradigm, where users are directly subject to the platform's algorithms. This paradigm places users in a vulnerable position, such as lack of control over their recommendation results, potentially being manipulated by the platform's recommendation algorithms, being trapped in echo chambers, or lack of personalization for those less active users due to the active users' dominance of the recommendation algorithm.

Firstly, the majority of recommendation models (Cheng et al., 2016; Kang & McAuley, 2018; Hidasi et al., 2015) are designed to optimize the commercial objectives of the platforms, such as increasing user clicks or conversion rates in e-commerce. This often results in users losing sight of their actual needs due to the algorithmic manipulation (Aguirre et al., 2015; Edizel et al., 2020; Grisse, 2023). Secondly, although recommendation models aims at offering personalized services, they are primarily

---

[1] https://anonymous.4open.science/r/iAgent-675C

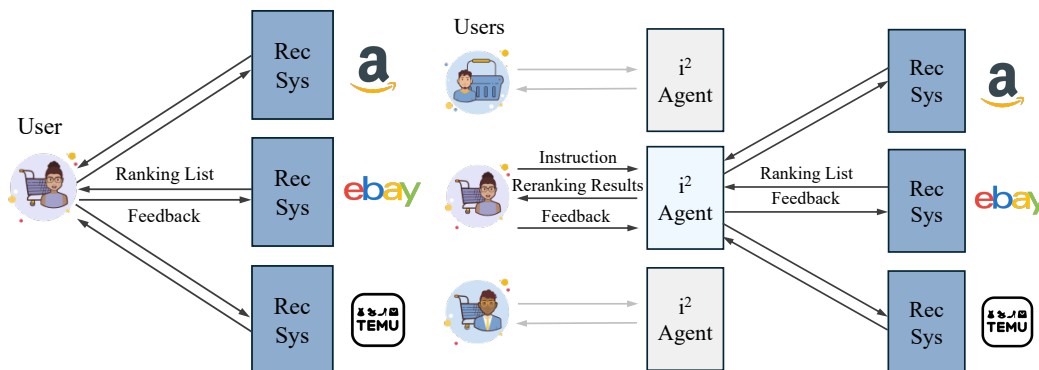

Figure 1: (a) Previous recommendation ecosystem primarily focused on designing sophisticated models to enhance the ranking performance so as to increase platform's benefit. However, they overlooked the user's proactive instructions and put users under the direct control of recommender systems. (b) In contrast, we build an individual instruction-aware agent for each user, which generates re-ranking results based on the user's active instructions. The agent's memory component is influenced solely by the individual user, providing an individualized personal service.

optimized based on data from all users, paying insufficient attention to individual preferences and unique interests (Patro et al., 2020; Li et al., 2021; Ge et al., 2022a). As a consequence of these shortcomings, users often fall into the echo chamber effects (Ge et al., 2020; Chitra & Musco, 2020; Bakshy et al., 2015), where algorithms reinforce user's existing interests or beliefs through repeated recommendation of homogeneous items, leading to a lack of diversity in recommended contents. Furthermore, the models tend to be biased towards advantaged (active) users, neglecting the interests of disadvantaged (less-active) users, resulting in a lack of personalization for some users.

To tackle these issues, researchers have approached the problem from various perspectives. On one hand, efforts are made to better understand user interests, such as using user's explicit feedback to improve the model performance and explanation (Zhang et al., 2014; Xie et al., 2021) or allowing users to better express their needs through conversational recommender sysetms (CRS) (Gao et al., 2021; Zhang et al., 2018). On the other hand, comprehensive models are developed to infer user interests from various dimensions, such as capturing user's diverse interests based on multi-behavior and multi-interest modeling (Zhou et al., 2018; 2019; Li et al., 2019). Most recently, language-based agents are utilized to mock the user behaviors and explore the user interests (Zhang et al., 2024b;a).

However, the two challenges remain insufficiently addressed due to the reliance on modeling user interests across all users' data and the focus on platform-side optimization. *To address these limitations, we propose a new user-agent-platform paradigm, where agent serves as the protective shield between user and recommender system that enables indirect exposure.* Our contributions are three-fold:

○ *New Datasets and Problem*: To provide benchmarks for the new user-agent-platform paradigm, we construct four recommendation datasets with user-driven instructions, referred to as INSTRUCTREC, constructed from existing datasets such as Amazon, Goodreads, and Yelp. Building on this, we propose an Instruction-aware Agent (iAgent), designed to learn user interests from the provided free-text instructions while leveraging external knowledge to act as a domain-specific expert. Unlike the instructions in CRS (Sun & Zhang, 2018) and Webshop (Yao et al., 2022), the free-text instructions in INSTRUCTREC allow users to flexibly express their requirements beyond just product attributes.

○ *Agent Learning from Individual Feedback*: We design Individual Instruction-aware Agent ($i^2$Agent), incorporating a dynamic memory mechanism with a profile generator and dynamic extractor to further explore user interests and learn from user's individual feedback. The profile generator constructs and maintains a user-specific profile by leveraging historical information and feedback. The dynamic extractor captures evolving profiles and interests based on the user's real-time instructions. Different from existing recommendation models, $i^2$Agent is optimized specifically for individual users and is not influenced by the interests or behaviors of other users, protecting the interests of less-active users.

○ *Empirical Results*: Empirical experiments on four datasets demonstrate that our $i^2$Agent consistently outperforms state-of-the-art approaches, achieving an improvement of up to 16.6% on average

Table 1: Difference between previous recommendation models and our model.

| Model | Instruction Awareness | Instruction Type | Dialogue Interaction | Dynamic Interest | Learning from Feedback | External Knowledge |
|-------|----------------------|------------------|---------------------|------------------|----------------------|--------------------|
| SR | ✗ | N/A | N/A | ✗ | ✗ | ✗ |
| CRS | ✓ | Fixed | Multiple Turns | ✓ | ✗ | ✗ |
| RecAgent | ✗ | N/A | N/A | ✗ | ✗ | ✓ |
| Ours | ✓ | Flexible | 0, 1, or Multiple Turns | ✓ | ✓ | ✓ |

across standard ranking metrics. Besides, we evaluate the impact of the echo chamber effect as well as the performance of both active and less-active users separately. From the overall empirical results, it validates that our proposed $i^2$Agent serve as a shield between user and recommender systems.

## 2 TASK DEFINITIONS AND COMPARISIONS

**Sequential Recommendation.** Consider a set of users $U$ and a set of items $I$. Each user's historical interactions are represented by a sequence $S_u = [s_1, \ldots, s_i, \ldots, s_T]$, where $s_i \in I$ and $T$ is the length of the sequence. The goal of sequential recommendation is to predict the next item $s_{T+1}$ that the user $u$ is likely to interact with, based on their past interactions $S_u$ (Hidasi et al., 2015; Kang & McAuley, 2018; Sun et al., 2019; Geng et al., 2022). Formally, this involves estimating the probability distribution over the items for the next interaction:

$$\hat{i} = \arg\max_{i \in I} P(s_{T+1} = i \mid S_u; \psi). \tag{1}$$

where $\psi$ is the model's parameters. Recent work on recommendation agents (Zhang et al., 2024b; Wang et al., 2023; Zhang et al., 2024a) has leveraged large language models (LLMs) to simulate user behavior by prompting them with plain text descriptions of user history and learn from the external knowledge via tool usage. Despite the shift to a language-based framework, it shares the same optimization objective as the traditional sequential recommendation.

**Conversational Recommendation.** Traditional conversational recommendation system (Sun & Zhang, 2018; Zhang et al., 2018) analyzes the user's intention via the multiple turn dialogue and consider historical information to achieve personalized recommendation. Mathematically, the recommendation model part[2] can be summarized as:

$$\hat{i} = \arg\max_{i \in I} P(s_{T+1} = i \mid S_u, H_u; \psi). \tag{2}$$

where $H_u = [h_1, ..., h_R]$ represents multiple historical dialogues of a user, $R$ represents the number of dialogues and $\psi$ is the model's parameters.

**Our Task.** Unlike sequential and conversational recommendation, our task focuses on learning from user's instructions to build an agentic shield between user and recommender system and meanwhile provide personalized recommendations for users. Mathematically, this can be summarized as follows:

$$\hat{i} = \arg\max_{i \in I} P(s_{T+1} = i \mid S_u, \Omega_u, E; \psi_u). \tag{3}$$

where $\Omega_u$ represents the user's instructions, and $\psi_u$ denotes the user-specific model parameters. $E$ represents the external environment, which can supply real-time information to the agent.

In Table 1, we highlight the key differences between previous recommendation models and our proposed model. Unlike existing recommendation models, our approach conducts an in-depth analysis of users' instructions and learns from individual feedback. Additionally, leveraging the power of LLMs, our model supports a highly flexible range of instructions and dialogues.

## 3 METHODOLOGY

In this part, we firstly introduce the naive solution iAgent based on INSTRUCTREC, which can learn the intention from the user instruction. Next, we introduce our $i^2$Agent equipped with individual dynamic memory. The workflow of models are shown in Fig. 2. All the prompt templates used in iAgent and $i^2$Agent and examples of responses are provided in Appendix B.

---

[2]The conversational model part is omitted for concise.

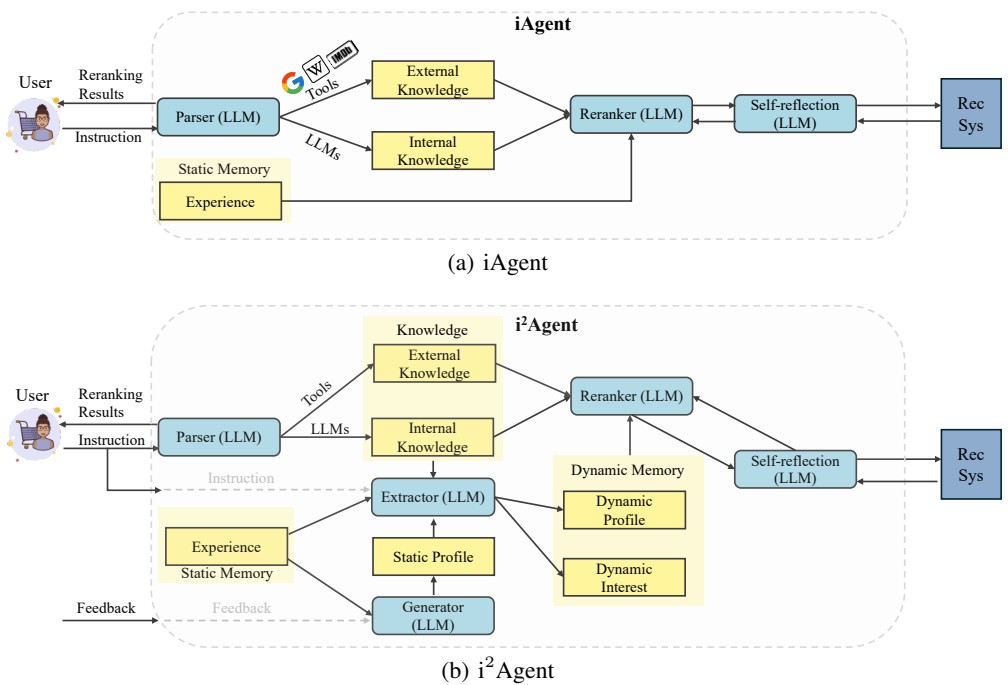

(a) iAgent

(b) i²Agent

Figure 2: Workflow of our proposed agents. (a) iAgent explores the relative knowledge under the user's instruction and provides the reranking results refined by the self-reflection mechanism. (b) i²Agent designs the dynamic memory mechanism to improve the personalized ability of iAgent.

### 3.1 IAGENT

**Parser.** The user's instructions encompass both direct lower-level demands and hidden higher-order preferences. Addressing these higher-order preferences requires agents to be equipped with relevant knowledge, transforming them into domain-specific experts that serve the user. Domain-specific experts use their professional knowledge to recognize differences between products, such as parameterized variations, and connect these distinctions to the user's expressed needs. The parser model is built upon a large language model (LLM), represented by $M_p$, which is specifically prompted to generate internal knowledge and decide whether to use external tools to extract knowledge from the open world based on the given instruction. In the first step, we concatenate the instruction $X_I$ with the parser's prompt template $P_{tp}$ and prompt the LLM to output the related internal knowledge $X_{IK}$ about the instruction. This step also involves deciding whether to use external tools $O_T$ and generating the instruction keywords $X_{KW}$. For example, in the book domain, this may include understanding each book's theme, types of storylines, and other related aspects. Next, if the parser $M_p$ decides to use external tools, the instruction keywords $X_{KW}$ and the potential tool options $O_T$ are utilized to explore the external knowledge $X_{EK}$.

$$O_T, X_{KW}, X_{IK} \leftarrow M_p(X_I \| P_{tp}); \quad X_{EK} \leftarrow M_p(O_T \| X_{KW}) \quad (4)$$

**Reranker.** After obtaining the instruction-related knowledge, the reranker, denoted by the LLM-based model $M_r$, reranks the initial ranking list $\mathcal{R}$ from the recommender platform. In addition to the generated knowledge $X_{IK}$ and $X_{EK}$, we incorporate the user's historical sequential information $X_{SU}$, which serves as a static memory of the user. Similarly, the textual information $X_{Item}$ of the items in the ranking list is also provided. Overall, the instruction-related knowledge, the textual information $X_{SU}$ and $X_{Item}$, along with the reranker's prompt template $P_{tr}$, are fed into the reranker $M_r$. Formally, this process can be written as follows:

$$\mathcal{R}^* \leftarrow M_r(X_{IK} \| X_{EK} \| X_{SU} \| X_{Item} \| P_{tr}) \quad (5)$$

where $\mathcal{R}^*$ is the reranked item lists and $X_{Item}$ includes the textual information (such as title and description) of the candidate items and item index from the initial ranking list $\mathcal{R}$.

**Self-reflection Mechanism.** Large language models output content in a generative manner, which can lead to hallucination problems (Huang et al., 2023a). To address this, we designed a self-reflection

mechanism to verify the content of the re-ranked item list. Specifically, we compare the elements between the reranking list and the previous one. If no differences are found, the results are directly output. However, if discrepancies are detected, the self-reflection module invokes the reranker to regenerate the reranking list, adding a prompt $P_{sr}$ to ensure alignment with the original ranked list. The mathematical formulation remains the same as in Eq. 5, with the prompt $P_{tr}$ replaced by $P_{sr}$.

### 3.2 I²AGENT

Although our basic framework iAgent can explore knowledge based on the user's instructions, it fails to effectively model the dynamic interests within the instructions and cannot learn from user feedback.. To address this, we design a profile generator to build user's personal profile that learns from the user feedback and a dynamic extractor to extract dynamic interest and build dynamic profile according to the instruction. Unlike existing recommendation models, i²Agent is uniquely optimized for individual users, remaining unaffected by the behaviors of other users.

**Profile Generator.** In our profile generator, we simulate the training process of a neural network by first feeding training data pairs into the generator, followed by presenting the ground truth interacted item and the corresponding reviews. Consider a user with a sequence of interactions, where the most recent interacted item is selected as the positive sample, and a negative item is randomly selected from the non-interacted items. The sampled pair, along with their corresponding textual information, are combined and fed into the generator $M_{ge}$, which selects one item from the two as the recommended item for the user. Moreover, the user's static memory $X_{SU}$ and the rank prompt template $P_{pr1}$ are also input into the model. Formally, this process can be expressed as:

$$X_G^T \leftarrow M_{ge}(X_{SU} \parallel X_i^+ \parallel X_i^- \parallel \mathcal{F}^{T-1} \parallel P_{pr1}) \tag{6}$$

where $X_i^+$ and $X_i^-$ represent the textual information of the positive and negative samples, respectively, and $\mathcal{F}^{T-1}$ denotes the user's profile in the previous round of interaction. $X_G^T$ is the recommended item generated by $M_{ge}$. $T$ represents the round of feedback update iterations. Then, we incorporate user feedback to further update the user's profile in this round. This feedback includes the groundtruth interacted item and any optional reviews. The generator $M_{ge}$ integrates this information as follows:

$$\mathcal{F}^T \leftarrow M_{ge}(\mathcal{F}^{T-1} \parallel X_i^{+*} \parallel X_G^T \parallel P_{pr2}) \tag{7}$$

where $X_i^{+*}$ contains the positive sample's textual information augmented with feedback data, and $P_{pr2}$ is the corresponding prompt template.

**Dynamic Extractor.** Similar to the attention mechanism (Vaswani, 2017), we propose a dynamic extractor to extract instruction-relative information based on the instruction. We prompt the extractor ($M_e$) to extract the dynamic interest from the static memory of user historical information $X_{SU}$ and the generated profile $\mathcal{F}_T$ according to the instruction $X_I$ and the generated instruction-related knowledge $X_{IK}$ and $X_{EK}$. It can be formulated as:

$$\mathcal{F}_d^T, X_{DU} \leftarrow M_e(\mathcal{F}^T \parallel X_{SU} \parallel X_I \parallel X_{IK} \parallel X_{EK} \parallel P_e) \tag{8}$$

where $\mathcal{F}_d^T$ and $X_{DU}$ represents the dynamic profile and dynamic interest, respectively. These two components form the dynamic memory. $P_e$ is the prompt template.

**Reranker.** After constructing the dynamic memory of a user, the reranker utilizes the information to generate the reranked results. Similar to Eq. 5, it can be expressed as:

$$\mathcal{R}^* \leftarrow M_r(X_{IK} \parallel X_{EK} \parallel X_{SU} \parallel \mathcal{F}_d^T \parallel X_{DU} \parallel X_{Item} \parallel P_{tr}^*) \tag{9}$$

where $P_{tr}^*$ represents the prompt template for the reranker in i²Agent. Besides, a self-reflection mechanism is also implemented to ensure consistent results, using the same inputs as the reranker, except for the prompt template.

## 4 EMPIRICAL EVALUATION

In this section, we present extensive experiments to demonstrate the effectiveness of iAgent and i²Agent, aiming to answer the following four research questions (**RQs**).

Table 2: Statistics of the INSTRUCTREC dataset: $|\mathcal{U}|$, $|\mathcal{V}|$, and $|\mathcal{E}|$ represent the number of users, items, and interactions, respectively. $\#|X_I|$ denotes the average token length of user instructions, while $\#|S_U|$ represents the average token length of the user's static memory.

| Dataset | $|\mathcal{U}|$ | $|\mathcal{V}|$ | $|\mathcal{E}|$ | Density | $\#|X_I|$ | $\#|S_U|$ |
|---|---|---|---|---|---|---|
| INSTRUCTREC - Amazon Book | 7,377 | 120,925 | 207,759 | 0.023% | 164 | 1276 |
| INSTRUCTREC -Amazon Movietv | 5,649 | 28,987 | 79,737 | 0.049% | 40 | 726 |
| INSTRUCTREC - Goodreads | 11,734 | 57,364 | 618,330 | 0.092% | 41 | 2827 |
| INSTRUCTREC - Yelp | 2,950 | 31,636 | 63,142 | 0.068% | 40 | 1976 |

• **RQ1**: How does the performance of iAgent and i²Agent compare to state-of-the-art baselines across various datasets?

• **RQ2**: Can our method mitigate the echo chamber effect by helping users filter out unwanted ads and recommending more diverse items, rather than just recommending popular ones?

• **RQ3**: How well does our method perform for both active and less-active user groups?

• **RQ4**: Are the proposed reranker and self-reflection mechanism effective in practice?

## 4.1 EXPERIMENT SETUP

**Dataset.** Given the absence of a recommendation dataset that includes proactive user instructions in the user-agent-platform paradigm, we construct INSTRUCTREC datasets using existing recommendation datasets, including Amazon (Ni et al., 2019), Yelp[3], and Goodreads (Wan et al., 2019). These datasets provide textual information such as item titles, descriptions, and reviews. We eliminate users and items that have fewer than 5 associated actions to ensure sufficient data density. For each interaction, we generate the instruction for this interaction based on the corresponding user review and filter through a post-processing verification mechanism. To further enhance the linguistic diversity of the instructions, we assign a persona to each user. More details are in the following.

*Instruction Generator*: Initially, we manually annotate several instruction-review pairs, providing few-shot examples for LLMs to facilitate in-context learning. These few-shot examples, along with reviews paired with a random persona from Persona Hub (Chan et al., 2024), are then fed into the LLM[4] to generate instructions. To ensure that the few-shot examples remain dynamic, we create a list to store the instruction-review pairs and allow the LLM to decide whether a newly generated instruction should be included as an example. Examples of the annotated instruction-review pairs, generated instructions, and the data construction processes can be found in Appendix B.3.

*Instruction Cleaner*: To prevent data leakage from the reviews, we test if or not the LLM can recover the item from the generated instruction. More specifically, given the instruction, we employ the LLM to choose between the ground-truth item and a randomly selected negative item. The LLM generates a certainty score based on the instruction and the item's textual information. Based on the result, we retain all of those instructions for which the LLM cannot infer the ground-truth item, and also keep an equal number of correctly inferred instructions that has low certainty scores. Statistical analysis of INSTRUCTREC dataset is in Table 2. For the filtered instructions and the retained instructions, we show some examples in Appendix B.3.2.

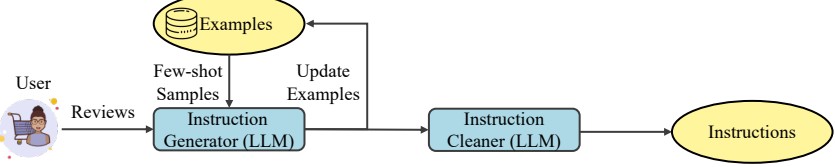

Figure 3: The overview of our INSTRUCTREC dataset construction.

**Evaluation Protocol.** We randomly sample 9 negative items with one true item to make the candidate ranking list. Following the data split in sequential recommendation (Kang & McAuley, 2018), the

---

[3]https://www.kaggle.com/datasets/yelp-dataset/yelp-dataset/versions
[4]We use GPT-4o-mini for data generation.

Table 3: Evaluation results (%) of the ranking metric (↑) on the INSTRUCTREC. We highlight the methods with the **first**, **second** and **third** best performances.

| Model | INSTRUCTREC − Amazon Book | | | | INSTRUCTREC − Amazon Movietv | | | |
|---|---|---|---|---|---|---|---|---|
| | **HR@1** | **HR@3** | **NDCG@3** | **MRR** | **HR@1** | **HR@3** | **NDCG@3** | **MRR** |
| GRU4Rec | 11.00 | 31.41 | 22.53 | 30.10 | 15.80 | 36.85 | 27.63 | 34.36 |
| BERT4Rec | 11.48 | 30.90 | 22.32 | 30.31 | 14.74 | 35.13 | 26.36 | 33.43 |
| SASRec | 11.08 | 31.34 | 22.42 | 30.15 | 34.52 | 49.71 | 43.18 | 48.06 |
| BM25 | 9.92 | 24.48 | 18.21 | 27.00 | 11.29 | 30.27 | 22.09 | 30.04 |
| BGE-Rerank | 25.36 | 45.90 | 37.11 | 42.84 | 25.44 | 47.48 | 38.02 | 43.28 |
| EasyRec | **30.70** | **48.87** | **41.09** | **46.14** | **34.96** | **61.30** | **50.15** | **52.98** |
| ToolRec | 10.56 | 30.60 | 21.88 | 29.77 | 13.84 | 35.67 | 26.20 | 33.21 |
| AgentCF | 14.24 | 34.16 | 25.55 | 32.77 | 25.90 | 49.82 | 39.64 | 44.23 |
| iAgent | **31.89** | **48.99** | **41.69** | **47.23** | **38.19** | **56.87** | **48.93** | **53.04** |
| i$^2$Agent | **35.11** | **53.51** | **45.64** | **50.28** | **46.43** | **65.77** | **57.67** | **60.43** |

most recent interaction is reserved for testing. The agent-based works, including ours, utilize all the interaction data except the most recent one to construct the agent's memory. For evaluation metric, we adopt the typical top-$N$ metrics hit rate (HR@$\{1, 3\}$), normalized discounted cumulative gain (NDCG@$\{3\}$) (Järvelin & Kekäläinen, 2002) and Mean Reciprocal Rank (MRR) (Sarwar et al., 2001). In addition to conventional ranking metrics, we conduct additional experiments to ensure that our iAgent/i$^2$Agent can act as a shield between users and the recommendation system. Specifically, we design evaluation metrics such as the percentage of filtered Ads items (FR@1,3,5,10) and popularity-weighted ranking metrics (P-HR@3 and P-MRR) to validate the mitigation of the echo chamber effect (Ge et al., 2020; Xu et al., 2022). We use freq$_i$ to denote the frequency of item $i$ in the dataset. Formally, these metrics are defined as:

$$\text{FR@k} = \begin{cases} 1, & \text{if } r_{Ads} > k, \\ 0, & \text{if } r_{Ads} \le k. \end{cases} \qquad \text{P-Rank} = (1 - \sigma\left(\text{freq}_i\right)) \cdot \text{Rank}. \qquad (10)$$

where $r_{Ads}$ denotes the position of Ads items in the re-ranked list, Rank represents ranking metrics such as HR, and $\sigma$ refers to the sigmoid function. The Ads items is randomly selected from a different data domain. For example, to simulate the Ads items in INSTRUCTREC - Amazon Book, we select Ads items from the data in INSTRUCTREC - Amazon Movietv, to test if the agent is able to demote an irrelevant item even if the item is already added into the ranking list by the recommender system. Additionally, we report the performance for both active and less-active users separately (Li et al., 2021). We also analyze the probability of changes in the top-ranked items after reranking. To further assess the effectiveness of our self-reflection mechanism, we report the occurrence rate of hallucination. For all evaluation metrics in our experiments, higher values indicate better performance.

**Baselines.** We compare our method with three classes of baselines: (1) Sequential recommendation methods, i.e., BERT4Rec (Sun et al., 2019), GRU4Rec (Hidasi et al., 2015) and SASRec (Kang & McAuley, 2018). (2) Instruction-aware methods, i.e., BM25 (Robertson et al., 2009), BGE-Rerank (Xiao et al., 2023) and EasyRec (Ren & Huang, 2024). (3) Recommendation agents, i.e., ToolRec (Zhao et al., 2024) and AgentCF (Zhang et al., 2024b). Detailed implementation and introduction of baselines are in Appendix A.

## 4.2 Performance Comparison

**Main Results.** (**RQ1**) Tables 3 and 4 present the experimental results across four datasets using different evaluation metrics. By incorporating instruction knowledge into the model, the instruction-aware baselines outperform traditional recommendation agent methods. Benefiting from the alignment with collaborative filtering and natural language information, EasyRec pretraining on several Amazon datasets achieves the second-best results, trailing only our iAgent. Our i$^2$Agent outperforms the second-best baseline, EasyRec, with the averagely 16.6% improvement. This improvement is partly attributed to the parser component, which learns instruction-aware knowledge, enabling the reranker to better understand the user's intentions. Meanwhile, our proposed dynamic memory component leverages user feedback to construct a more accurate user profile and dynamically extract interests from historical data based on the instruction.

Table 4: Evaluation results (%) of the ranking metric (↑) on INSTRUCTREC.

| Model | INSTRUCTREC – Goodreads | | | | INSTRUCTREC – Yelp | | | |
|---|---|---|---|---|---|---|---|---|
| | HR@1 | HR@3 | NDCG@3 | MRR | HR@1 | HR@3 | NDCG@3 | MRR |
| GRU4Rec | 15.36 | 39.52 | 29.08 | 35.41 | 10.94 | 30.67 | 21.88 | 29.70 |
| BERT4Rec | 12.70 | 34.69 | 25.02 | 32.32 | 10.99 | 31.02 | 22.32 | 30.05 |
| SASRec | 18.52 | 41.24 | 31.47 | 37.60 | 12.59 | 31.09 | 22.65 | 30.15 |
| BM25 | 14.25 | 40.34 | 29.01 | 35.40 | 12.85 | 33.08 | 24.34 | 31.85 |
| BGE-Rerank | 17.26 | 40.82 | 30.60 | 36.97 | **33.05** | 55.29 | 45.70 | **49.90** |
| EasyRec | 13.94 | 35.38 | 26.11 | 33.27 | 32.41 | **56.31** | **46.04** | 49.86 |
| ToolRec | 19.06 | 42.79 | 32.61 | 38.44 | 12.07 | 30.92 | 22.83 | 30.21 |
| AgentCF | **21.61** | **46.09** | **35.60** | **40.96** | 13.36 | 34.83 | 25.66 | 32.61 |
| iAgent | **23.56** | **47.01** | **36.98** | **42.19** | **37.40** | **56.33** | **48.28** | **52.42** |
| i$^2$Agent | **30.97** | **56.69** | **45.76** | **49.14** | **39.22** | **57.92** | **49.96** | **53.78** |

Table 5: Evaluation of the echo chamber effects (%) (↑) on INSTRUCTREC.

| Model | INSTRUCTREC – Amazon Book | | | | INSTRUCTREC – Yelp | | | |
|---|---|---|---|---|---|---|---|---|
| | FR@1 | FR@3 | P-HR@3 | P-MRR | FR@1 | FR@3 | P-HR@3 | P-MRR |
| EasyRec | **68.41** | **64.32** | **59.28** | **56.09** | 76.45 | 66.50 | 61.05 | 56.85 |
| ToolRec | 70.13 | 66.61 | 36.74 | 35.80 | 72.64 | 63.64 | 32.50 | 32.73 |
| AgentCF | 58.02 | 50.04 | 41.10 | 39.42 | 71.30 | 64.15 | 38.46 | 36.44 |
| iAgent | **71.98** | **67.82** | **59.51** | **57.32** | **78.24** | **69.71** | **62.74** | **58.76** |
| i$^2$Agent | **77.15** | **70.15** | **64.70** | **60.87** | **87.69** | **84.20** | **64.48** | **60.20** |

Table 6: The performance (%) of active and less-active users on INSTRUCTREC - Amazon book.

| Model | Less-Active Users | | | | Active Users | | | |
|---|---|---|---|---|---|---|---|---|
| | HR@1 | HR@3 | NDCG@3 | MRR | HR@1 | HR@3 | NDCG@3 | MRR |
| EasyRec | **32.93** | **51.07** | **43.32** | **48.04** | 28.71 | 47.64 | 39.53 | 44.61 |
| ToolRec | 10.57 | 30.86 | 22.01 | 29.88 | 10.04 | 31.73 | 22.32 | 29.54 |
| AgentCF | 14.79 | 35.00 | 26.26 | 33.35 | 14.87 | 34.37 | 25.93 | 33.24 |
| iAgent | **34.07** | **50.79** | **43.67** | **49.00** | **29.96** | **47.73** | **40.14** | **45.71** |
| i$^2$Agent | **37.92** | **55.75** | **47.84** | **52.11** | **33.27** | **51.74** | **43.81** | **48.67** |

**Echo Chamber Effect. (RQ2)** We also report the experimental results evaluating the echo chamber effect in Table 5. Ads items are randomly inserted into the candidate ranking list from other domains to simulate advertising scenarios that users may have encountered. To mitigate position bias in LLMs (Liu et al., 2024b), Ads items are added randomly within the candidate list positions. i$^2$Agent accurately identifies users' instructions and extracts knowledge about their underlying needs, thereby effectively removing undesired Ads. Benefitting from not being trained in a purely data-driven manner and constructing user profiles based on their feedback, our i$^2$Agent also recommends more diverse items to users (both active and less-active items), instead of focusing solely on popular items, and meanwhile improves the overall recommendation performance. Drawing from these experimental results, we conclude that our i$^2$Agent can mitigate the echo chamber effect and act as a protective shield for users. Due to the page limitation, we provide full experiment results in Appendix A.3.1.

**Protect Less-Active Users. (RQ3)** We define the top 20% of users as active, with the remaining 80% classified as less-active (Li et al., 2021; Xu et al., 2023). Since our data is sampled and filtered using a 10-core process, most users exhibit rich behavioral patterns. Consequently, active users tend to show poorer performance compared to less-active users, largely due to the decline in LLM performance with longer texts (Liu et al., 2024a). As illustrated in Table 6, our i$^2$Agent enhances the performance for both active and less-active users. For less-active users, we construct individual profiles based on their feedback, ensuring that these profiles are not influenced by other users. The experimental results demonstrate that our dynamic memory mechanism offers personalized services tailored to each user individually. Detailed implementation and introduction of baselines are in Appendix A.3.2.

**Model Study. (RQ4)** First, we analyze the impact of our self-reflection mechanism on the LLM's hallucination rate. When implementing ToolRec (Zhao et al., 2024) and AgentCF (Zhang et al.,

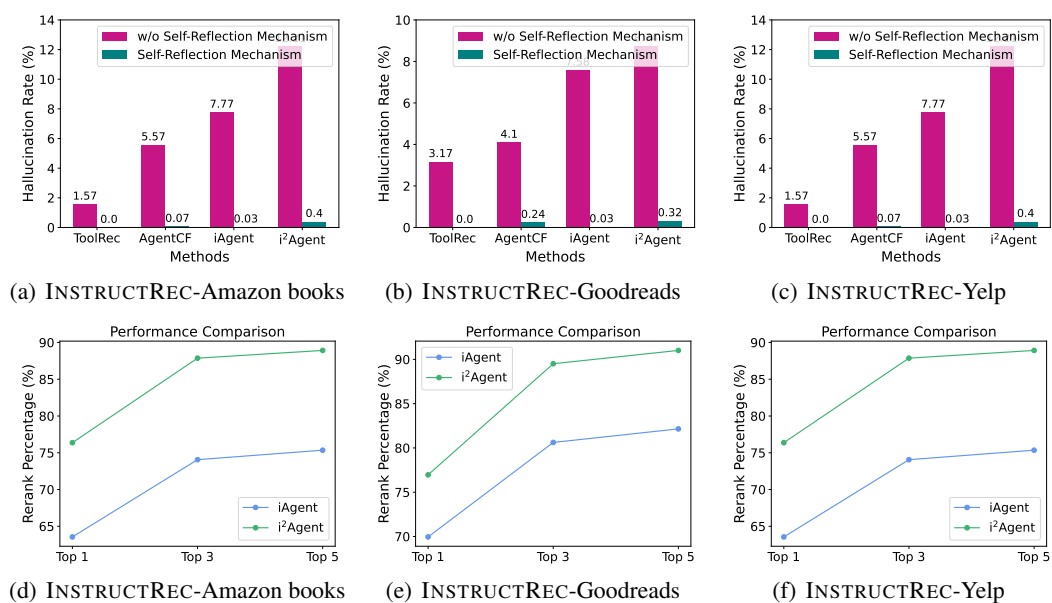

Figure 4: The first row presents the hallucination rate with and without the self-reflection mechanism, while the second row illustrates the probability of changes in the ranking list after our reranker.

2024b), we applied the self-reflection mechanism to improve the accuracy of the reranking list. As shown in Fig. 4, the self-reflection mechanism reduces the hallucination rate by at least 20-fold. In this mechanism, we prompt the LLM to generate the reranking list based on the initial ranking list. However, i²Agent exhibits the highest error rate, as the longer text sequence causes the LLM to lose some information from the original ranking list. Based on the experimental results, we can safely conclude that our self-reflection mechanism effectively alleviates LLM-induced hallucinations.

Next, we examine the re-ranking ratio across our models. We compare whether the elements in the ranking list change before and after reranking, focusing on the top@{1,3,5} positions. If any element changes position, it is considered a rerank. The results indicate that changes occur almost every time during reranking, suggesting that our agent is consistently performing personalized reranking on the list generated by the recommender platform.

## 5 RELATED WORK

### 5.1 RECOMMENDER SYSTEM

Sequential recommendation models (Hidasi et al., 2015; Sun et al., 2019; Kang & McAuley, 2018) primarily focus on developing temporal encoders to capture both short- and long-term user interests. For instance, SASRec (Kang & McAuley, 2018) leverages an attention mechanism to capture long-term semantics, while BERT4Rec (Sun et al., 2019) uses a bidirectional encoder with a masked item training objective. In the context of embracing large language models, generative recommenders (Geng et al., 2022; Zhai et al., 2024) treat item indices as tokens and predict them in a generative manner. Meanwhile, LLMs (Li et al., 2023; Xu et al., 2024) are utilized to play as a sequential embedding extractor to improve the recommendation performance. In our framework design, all recommendation models can be considered as components of the tools.

Before large language model become popular, conversational recommendation system (CRS) (Sun & Zhang, 2018; Zhang et al., 2018; Qu et al., 2019) aims at designing better dialogue understanding models or incorporating reinforcement learning for multiple dialogues answering. Due to the capacity of the conventional language model, it lose the flexibility of the dialogue including the dialogue format and number of turns. To resolve this problem, some researchers (Friedman et al., 2023; Feng et al., 2023) leverage the power of LLM to better understand the intention of user.

The echo chamber effect occurs when individuals are exposed only to information and opinions that reinforce their existing beliefs within their social networks (Bakshy et al., 2015; Chitra & Musco, 2020;

Garimella et al., 2018), leading to a lack of diverse perspectives and increased polarization (Aslay et al., 2018; Kaminskas & Bridge, 2016; Kunaver & Požrl, 2017). In the context of recommender systems, researchers have begun to study echo chambers and feedback loops (Ge et al., 2020; Xu et al., 2022; Chaney et al., 2018; Jiang et al., 2019; Möller et al., 2020; Kalimeris et al., 2021). Kalimeris et al. (Kalimeris et al., 2021) propose a matrix factorization-based recommender system with a theoretical framework for modeling dynamic user interests, while $\partial$CCF (Chitra & Musco, 2020) employs counterfactual reasoning to mitigate echo chambers.

## 5.2 Personal Language-based Agent

In the early stages, some researchers (Zhang, 2018; Park et al., 2023; Shanahan et al., 2023) in the NLP field developed dialogue agents with personas to enhance dialogue quality. Language models (Park et al., 2023) are prompted with role descriptions to simulate realistic interactions by storing experiences, synthesizing memories, and dynamically planning actions, resulting in believable individual and social behaviors within interactive environments. WebShop (Yao et al., 2022) attempts to understand product attributes from human-provided text instructions using reinforcement learning and imitation learning. Similar to traditional conversational recommender systems (CRS) (Zhang et al., 2018), it is impractical for users to describe each product attribute every time. With the advancement of large language models (such as GPTs (Achiam et al., 2023)), many researchers (Gur et al., 2023; Deng et al., 2024; Xie et al., 2024) have begun designing domain-specific agents that integrate various tool learning and memory mechanisms.

More recently, recommendation agents (RecAgent) (Zhao et al., 2024; Wang et al., 2023; Zhang et al., 2024a;b; Wang et al., 2024; Huang et al., 2023b) have been developed to simulate user behaviors and predict user-item interactions. A common design feature among these agents is the use of historical interaction information as user memory (Zhao et al., 2024; Wang et al., 2023; Huang et al., 2023b), with LLMs utilized to generate the ranking results. Unlike platform-side RecAgents, iAgent and i$^2$Agent are the first to operate on the user side, generating re-ranking results based on user instructions and individual memory, unaffected by the influence of advantaged users.

## 6 Conclusion and Future Direction

In this work, we first establish an instruction-aware recommendation benchmark and design a straightforward instruction-aware agent (iAgent) to analyze user instructions and integrate relevant and comprehensive knowledge. Moreover, to enhance the agent's personalized abilities, we propose individual instruction-aware agent (i$^2$Agent), which incorporates a dynamic memory mechanism to learn from user's personal feedback and extracts the dynamic interests. In addition to these technical contributions, our work also presents unique and complementary avenues for future research.

**More Effective Reranker.** In this version of iAgent and i$^2$Agent, we construct a zero-shot reranker based on LLMs, such as GPT4-o-mini. Recently, several open-source LLMs (Gunter et al., 2024; Abdin et al., 2024; Team et al., 2024), typically containing fewer model parameters (2-3 billion), have demonstrated strong performance. It is feasible to fine-tune smaller LLMs to build a more effective reranker on our INSTRUCTREC dataset. Furthermore, existing advanced recommendation models (Zhai et al., 2024; Xu et al., 2024) can serve as tools for the agent to retrieve candidate items.

**Multi-step Feedback.** Although we have constructed various datasets rich in abundant instructions, the feedback for re-ranking results is limited to a single ground-truth item, lacking continuous, multi-step feedback on interactions between users and agents. Additionally, the feedback explanations from users are insufficient. If i$^2$Agent were deployed in a real-world environment, more comprehensive feedback could be collected, enabling the development of more interpretable agents for users.

**Mutual Learning.** This work builds an agent for users that makes decisions for users and collect feedback from users. The platform-side recommendation models can improve their performance by leveraging the feedback and explanations provided by agents on behalf of their users. Furthermore, recommendation agents (Zhao et al., 2024; Zhang et al., 2024b; Wang et al., 2023; Zhang et al., 2024a) can autonomously and iteratively improve through mutual learning with i$^2$Agent. Moreover, i$^2$Agent can serve as a reward function for RL-based recommendation models (Afsar et al., 2022; Zheng et al., 2018; Wang et al., 2020; Ge et al., 2022b), enhancing their performance.

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

# APPENDIX

## CONTENTS

# A   EXPERIMENT

## A.1   SOURCE DATASET

**Amazon Book/Movietv** [5] (Ni et al., 2019) The Amazon product dataset is a comprehensive repository of consumer reviews and associated metadata, encompassing 142.8 million reviews collected over an 18-year span from May 1996 to July 2014. For our experiments, we leverage two distinct subsets: "Books" and "Movies and TV." Each dataset includes anonymized user and item identifiers, along with user-provided ratings on a 1-5 scale and corresponding textual reviews. Furthermore, rich product metadata is incorporated, such as detailed descriptions, categorical classifications, pricing information, and brand data. This multifaceted dataset provides a fertile ground for both collaborative filtering and content-based recommendation approaches, where the interplay between user behavior, product attributes, and textual feedback can be modeled to advance the state of recommendation systems.

**Goodreads.** [6] (Wan et al., 2019) The Goodreads dataset is derived from one of the largest online platforms dedicated to book reviews, offering user-generated ratings, reviews, and a variety of associated metadata. Each user in the dataset is represented by an anonymized identifier, with interactions including rating and reviewing a broad selection of books. The books are identified through International Standard Book Numbers (ISBNs) and accompanied by an extensive set of metadata, including title, author, publication year, and genre classifications. This data is especially valuable for the development of content-aware recommendation models, where leveraging the contextual features of both user interactions and book attributes can enhance predictive accuracy. The textual reviews, in particular, provide a rich source of natural language data, capturing nuanced user feedback that can be further utilized in sentiment analysis, opinion mining, and advanced NLP tasks. Ratings, similarly to the Amazon dataset, are presented on a 1-5 scale, providing a consistent metric for comparative analysis across different datasets.

**Yelp.** [7] The Yelp dataset contains over 67,000 reviews focused on businesses, particularly restaurants, from three major English-speaking cities, sourced from the popular Yelp platform. The dataset includes detailed metadata on both businesses and user interactions. Each business is uniquely identified and linked to comprehensive metadata, including its name, geographic location, category (e.g., restaurant, bar, or retail establishment), and additional attributes such as parking availability and reservation policies. This data is invaluable for context-aware recommendation systems, where business features and user feedback intersect to inform personalized recommendations. Anonymized user IDs track user interactions, with additional features such as the number of reviews written, average rating, and social features (e.g., "friends," "useful votes"). Yelp's textual reviews provide a rich dataset for natural language processing, where the diverse nature of user opinions, combined with structured metadata, offers a robust framework for evaluating and improving context-aware recommendation models.

## A.2   COMPARED METHODS

### A.2.1   SEQUENTIAL RECOMMENDATION METHODS

For the sequential recommendation baselines, only item ID information was considered in the model. To optimize performance, we experimented with various hyperparameters. The embedding dimension was tested across $\{32, 64, 128\}$, while the hidden representation in the prediction head ranged from $\{8, 16, 32\}$. Additionally, the learning rate was evaluated with values of $\{1e^{-3}, 4e^{-3}, 1e^{-4}, 4e^{-4}\}$. The best results are reported based on the highest MRR metric on the validation set.

**GRU4Rec** (Hidasi et al., 2015) addresses the challenge of modeling sparse sequential data while adapting RNN models to recommender systems. The authors propose a new ranking loss function

---

[5] https://cseweb.ucsd.edu/~jmcauley/datasets/amazon_v2/
[6] https://mengtingwan.github.io/data/goodreads
[7] https://www.kaggle.com/datasets/yelp-dataset/yelp-dataset/versions

specifically designed for training these models. The PyTorch implementation of GRU4Rec is available at the URL[8].

**BERT4Rec** (Sun et al., 2019) introduces a bidirectional self-attention network to model user behavior sequences. To prevent information leakage and optimize training, it employs a Cloze objective to predict randomly masked items by considering both their left and right context. The PyTorch implementation of BERT4Rec can be found at the URL[9].

**SASRec** (Kang & McAuley, 2018) is a self-attention-based sequential model designed to balance model parsimony and complexity in recommendation systems. Using an attention mechanism, SASRec identifies relevant items in a user's action history and predicts the next item with relatively few actions, while also capturing long-term semantics, similar to RNNs. This allows SASRec to perform well on both sparse and denser datasets. The PyTorch implementation of SASRec is available at the URL[10].

### A.2.2 INSTRUCTION-AWARE METHODS

We treat the concatenated text of the instruction as the query, while each candidate item is represented by its various metadata (e.g., title, description), transformed into textual format. These textual representations of candidate items are treated as individual 'documents,' forming the document corpus that instruction-aware methods rank based on relevance to the query. By leveraging the semantic richness of both the query and item metadata, this approach enables a context-aware ranking system, prioritizing items according to their alignment with the user's intent and preferences as conveyed through the instruction.

**BM25.** (Robertson et al., 2009) BM25, a probabilistic ranking function, is a foundational method in information retrieval, widely used to rank documents based on their relevance to a given query. The core concept of BM25 is to measure the similarity between a query and a document by considering both the frequency of query terms within the document and the distribution of those terms across the entire document corpus. BM25 balances two key factors: term frequency, which reflects how often a query term appears in a document (assuming that higher frequency indicates greater relevance), and inverse document frequency, which assigns more weight to rarer terms in the dataset, as they carry greater informational value. The PyTorch implementation of BM25 is available at the URL[11].

**BGE-Rerank.** (Xiao et al., 2023) The BGE-Rerank model utilizes a cross-encoder architecture, where both the query and document are processed together as a single input to directly generate a relevance score. Unlike bi-encoder models, which create independent embeddings for the query and document before computing their similarity, the cross-encoder applies full attention over the entire input pair, capturing more fine-grained interactions. This approach leads to higher accuracy in estimating relevance. In our implementation, we use the BGE-Rerank model to reorder candidate documents based on the relevance score for each query-document pair. The PyTorch implementation of BGE-Rerank is available at the URL[12].

**EasyRec.** EasyRec (Ren & Huang, 2024) is a lightweight, highly efficient recommendation system based on large language models, shown through extensive evaluations to outperform many LLM-based methods in terms of accuracy. Central to its success is the use of contrastive learning, which effectively aligns semantic representations from textual data with collaborative filtering signals. This approach enables EasyRec to generalize robustly and adapt to new, unseen recommendation data. The model employs a bi-encoder architecture, where text embeddings for queries and documents are pre-computed independently. These embeddings are then used to calculate similarity scores, allowing for the reordering of candidate items based on relevance. The PyTorch implementation of EasyRec is available at the URL[13].

---

[8]https://github.com/hungpthanh/GRU4REC-pytorch
[9]https://github.com/jaywonchung/BERT4Rec-VAE-Pytorch
[10]https://github.com/pmixer/SASRec.pytorch
[11]https://github.com/dorianbrown/rank_bm25
[12]https://github.com/FlagOpen/FlagEmbedding/tree/master/FlagEmbedding/reranker
[13]https://github.com/HKUDS/EasyRec

Table 7: Evaluation effects (%) of the echo chamber (↑) on the INSTRUCTREC-Amazon Books. We highlight the methods with the **first**, **second** and **third** best performances.

| Model | Amazon Book | | | | Amazon Book | | | |
|---|---|---|---|---|---|---|---|---|
| | FR@1 | FR@3 | FR@5 | FR@10 | P-HR@1 | P-HR@3 | P-NDCG@3 | P-MRR |
| EasyRec | 68.41 | 64.32 | 60.30 | 0.03 | **37.60** | **59.28** | **50.00** | **56.09** |
| ToolRec | **70.13** | **66.61** | 62.41 | 0.00 | 12.63 | 36.74 | 26.24 | 35.80 |
| AgentCF | 58.02 | 50.04 | 41.32 | **0.06** | 17.00 | 41.10 | 30.68 | 39.42 |
| iAgent | 71.98 | 67.82 | 60.74 | 0.08 | 38.85 | 59.51 | 50.70 | 57.32 |
| i²Agent | 77.15 | 70.15 | 64.05 | 0.09 | 42.62 | 64.70 | 55.25 | 60.87 |

### A.2.3 RECOMMENDATION AGENTS

**ToolRec.** (Zhao et al., 2024) uses large language models (LLMs) to enhance recommendation systems by leveraging external tools. The methodology involves treating LLMs as surrogate users, who simulate user decision-making based on preferences and utilize attribute-oriented tools (such as rank and retrieval tools) to explore and refine item recommendations. This iterative process allows for a more fine-grained recommendation that aligns with users' preferences.

**AgentCF.** (Zhang et al., 2024b) AgentCF is an innovative approach that constructs both user and item agents, powered by LLMs, to simulate user-item interactions in recommender systems. These agents are equipped with memory modules designed to capture their intrinsic preferences and behavioral data. At its core, AgentCF facilitates autonomous interactions between user and item agents, enabling them to make decisions based on simulated preferences. A key feature of this framework is the collaborative reflection mechanism, through which agents continuously update their memory, thereby improving their capacity to model real-world user-item relationships over time.

To ensure a fair comparison and optimize computational efficiency, the number of memory-building rounds in AgentCF is set to 1, matching that of our i²Agent. In AgentCF's experiments, the dataset size is 100, which represents only around 0.1% of the size of our dataset. Moreover, to ensure the generated reranking list without hallucination, we also equipped ToolRec and AgentCF with our self-reflection mechanism.

## A.3 PERFORMANCE COMPARISON

### A.3.1 ECHO CHAMBER EFFECT

We also report the experimental results evaluating the echo chamber effect in Table 7, Table 8 and Table 9. Ads items are randomly inserted into the candidate ranking list from other domains to simulate advertising scenarios that users may have encountered. To mitigate position bias in LLMs (Liu et al., 2024b), Ads items are added randomly within the candidate list positions. i²Agent accurately identifies users' instructions and extracts knowledge about their underlying needs, thereby effectively removing undesired Ads. Benefitting from not being trained in a purely data-driven manner and constructing user profiles based on their feedback, our i²Agent also recommends more diverse items to users (both active and less-active items), instead of focusing solely on popular items, and meanwhile improves the overall recommendation performance. Drawing from these experimental results, we conclude that our i²Agent can mitigate the echo chamber effect and act as a protective shield for users.

### A.3.2 PROTECT LESS-ACTIVE USERS

We define the top 20% of users as active, with the remaining 80% classified as less-active (Li et al., 2021; Xu et al., 2023). Since our data is sampled and filtered using a 10-core process, most users exhibit rich behavioral patterns. Consequently, active users tend to show poorer performance compared to less-active users, largely due to the decline in LLM performance with longer texts (Liu et al., 2024a). As illustrated in Table 10, Table 11 and Table 12, our i²Agent enhances the performance for both active and less-active users. For less-active users, we construct individual profiles based on their feedback, ensuring that these profiles are not influenced by other users. The experimental results demonstrate that our dynamic memory mechanism offers personalized services tailored to each user individually.

Table 8: Evaluation effects (%) of the echo chamber (↑) on the INSTRUCTREC-Amazon Movietv and INSTRUCTREC-GoodReads. We highlight the methods with the **first**, **second** and **third** best performances.

| Model | Amazon Movietv | | | | GoodReads | | | |
|---|---|---|---|---|---|---|---|---|
| | P-HR@1 | P-HR@3 | P-NDCG@3 | P-MRR | P-HR@1 | P-HR@3 | P-NDCG@3 | P-MRR |
| EasyRec | **37.31** | 65.45 | 53.54 | 56.69 | 14.22 | 35.98 | 26.56 | 33.84 |
| ToolRec | 14.73 | 38.12 | 27.96 | 35.57 | 19.21 | 43.22 | 32.92 | 38.88 |
| AgentCF | 27.61 | 53.33 | 42.37 | 47.37 | **21.82** | **46.62** | **35.99** | **41.47** |
| iAgent | 40.50 | 60.71 | 52.11 | 56.61 | 23.75 | 47.50 | 37.34 | 42.68 |
| i²Agent | 49.51 | 70.47 | 61.67 | 64.69 | 31.22 | 57.33 | 46.23 | 49.71 |

Table 9: Evaluation effects (%) of the echo chamber (↑) on the INSTRUCTREC-Yelp. We highlight the methods with the **first**, **second** and **third** best performances.

| Model | Yelp | | | | Yelp | | | |
|---|---|---|---|---|---|---|---|---|
| | FR@1 | FR@3 | FR@5 | FR@10 | P-HR@1 | P-HR@3 | P-NDCG@3 | P-MRR |
| EasyRec | **76.45** | 66.50 | 57.16 | **0.05** | 37.18 | 61.05 | 52.51 | 56.85 |
| ToolRec | 72.64 | 63.64 | 53.29 | 0.00 | 12.40 | 32.50 | 23.88 | 32.73 |
| AgentCF | 71.30 | 64.15 | 52.01 | 0.02 | 14.73 | 38.46 | 28.33 | 36.44 |
| iAgent | 78.24 | 69.71 | 56.17 | 0.12 | 41.74 | 62.74 | 53.82 | 58.76 |
| i²Agent | 87.69 | 86.20 | 84.00 | 0.16 | 43.67 | 64.48 | 55.62 | 60.20 |

Table 10: The performance (%) of active and less-active users on INSTRUCTREC - Amazon Movietv. We highlight the methods with the **first**, **second** and **third** best performances.

| Model | Less-Active Users | | | | Active Users | | | |
|---|---|---|---|---|---|---|---|---|
| | HR@1 | HR@3 | NDCG@3 | MRR | HR@1 | HR@3 | NDCG@3 | MRR |
| EasyRec | **35.17** | 61.56 | 50.39 | 53.21 | 35.47 | 63.15 | 51.26 | 53.64 |
| ToolRec | 14.43 | 36.56 | 26.96 | 33.81 | 12.98 | 32.18 | 23.94 | 31.79 |
| AgentCF | 27.38 | 50.98 | 40.91 | 45.36 | 21.84 | 45.58 | 35.57 | 40.76 |
| iAgent | 39.36 | 57.85 | 49.98 | 53.96 | 34.95 | 55.19 | 46.88 | 51.02 |
| i²Agent | 47.32 | 66.64 | 58.57 | 61.22 | 44.71 | 64.99 | 56.60 | 59.30 |

Table 11: The performance (%) of active and less-active users on INSTRUCTREC - GoodReads. We highlight the methods with the **first**, **second** and **third** best performances.

| Model | Less-Active Users | | | | Active Users | | | |
|---|---|---|---|---|---|---|---|---|
| | HR@1 | HR@3 | NDCG@3 | MRR | HR@1 | HR@3 | NDCG@3 | MRR |
| EasyRec | 14.44 | 35.77 | 26.55 | 33.67 | 14.13 | 36.86 | 27.09 | 33.86 |
| ToolRec | 19.85 | 43.34 | 33.29 | 39.11 | 17.89 | 42.02 | 31.63 | 37.35 |
| AgentCF | 22.91 | 46.67 | 36.50 | 41.89 | 19.82 | 46.70 | 35.22 | 40.10 |
| iAgent | 24.57 | 48.12 | 38.00 | 43.04 | 22.62 | 46.96 | 36.64 | 41.70 |
| i²Agent | 32.67 | 58.08 | 47.28 | 50.46 | 29.76 | 55.39 | 44.56 | 48.19 |

Table 12: The performance (%) of active and less-active users on INSTRUCTREC - Yelp. We highlight the methods with the **first**, **second** and **third** best performances.

| Model | Less-Active Users | | | | Active Users | | | |
|---|---|---|---|---|---|---|---|---|
| | HR@1 | HR@3 | NDCG@3 | MRR | HR@1 | HR@3 | NDCG@3 | MRR |
| EasyRec | **32.83** | 56.50 | 46.29 | 50.13 | 30.17 | 50.87 | 42.03 | 47.16 |
| ToolRec | 11.79 | 31.21 | 22.88 | 30.14 | 14.21 | 32.42 | 24.66 | 32.11 |
| AgentCF | 13.11 | 34.72 | 25.50 | 32.46 | 13.22 | 36.41 | 26.45 | 32.89 |
| iAgent | 37.80 | 56.17 | 48.37 | 52.70 | 39.40 | 59.10 | 50.62 | 53.90 |
| i²Agent | 39.02 | 58.49 | 50.23 | 53.88 | 43.25 | 57.75 | 51.48 | 56.05 |

# B  PROMPT TEMPLATES AND EXAMPLES

All output messages are decoded in a JSON-structured format through the OpenAI service [14].

## B.1  PROMPT TEMPLATES AND EXAMPLES RESPONSE IN iAGENT

### B.1.1  PARSER

**With the Google Search Tools.**[15]

---

**The prompt template in Parser:** $P_{tp}$

```
Based on the following instruction, help me decide which
tools to use and generate the keywords for tool usage.
Please specify the types of descriptions that the recommended
items should include.  Do not directly recommend specific
items.  Don't use numerical numbering for the generated
content; you can use bullet points instead.
Instruction X_I:  I am on the hunt for a book that offers
a refreshing break from the daily grind, much like how
a filmmaker can offer a respite from the monotony of
car-centric society.  The narrative should be feel-good and
predictable, with a focus on athletic fashions for everyday
people, much like how cars have become a ubiquitous part
of modern life.  The book should offer an entertaining and
engaging escape from reality, much like how a filmmaker can
transport the audience to another world through the magic of
cinema.
```

**The generated query of Google Search API:** $X_{KW}$

```
feel-good books that offer an escape from reality and focus
on athletic fashion for everyday people.
```

**The response of Google Search API:**

```
Google Search Results:  X_EK
May 5, 2024 ...  She Wrote the First Great Perimenopause
Novel.  Miranda July is experimenting again | on the page
and in her life.  Mar 25, 2023 ...  Or even if it doesn't fit
this vibe, just any book that makes you happy.  Can't wait
for all of your recs!  :).  Jul 21, 2022 ...  ... feel good
about ourselves.  Life is a ...  The desire for a good and
meaningful life that inspires people to commit daily acts of
generosity.  Oct 3, 2022 ...  The philosopher Galen Strawson
believes that some people are simply more \episodic" than
others; they're fine living day to day, without regard ...
Nov 12, 2020 ...  Written by a westerner who came to Buddhism
later in life, I found it more accessible when I was new to
Buddhist books.  But yes, it helped me to ...  Jun 9, 2015
...  For all avid readers who have been self-medicating with
great books their entire lives, it comes as no surprise that
reading books can be good for you.  It was a way of life that
approximated a racial caste system (Woodward, 1974).  Jim
Crow laws and etiquette were aided by millions of material
objects that ...  Apr 5, 2020 ...  The Great English Seaside
Holiday in its heyday, and the beautiful dignity to be found
in everyday living, have rarely been captured more The rest
of the information is omitted....
```

---

[14]https://platform.openai.com/docs/guides/structured-outputs/introduction

[15]The Google Custom Search API operates on a pay-per-use pricing model. The JSON API, used to retrieve web or image search results, charges $5 per 1,000 queries. There is a limit of 10,000 queries per day.

**Without the Tools.** To provide a more reproducible experiment results, we do not utilize the Google Search API all the time, instead of utilizing LLMs to output relative knowledge. Beyond Goole Search API, perplexity search[16] is also a well-performance choice ignoring the price.

---

**The Prompt Template in Parser:** $P_{tp}$

```
Based on the following instruction, assist me in generating
relevant knowledge.
Please specify the types of descriptions that the recommended
items should include.
Do not directly recommend specific items.  Don't use
numerical numbering for the generated content; you can use
bullet points instead.
Instruction X_I:I am on the hunt for a book that offers
a refreshing break from the daily grind, much like how
a filmmaker can offer a respite from the monotony of
car-centric society.  The narrative should be feel-good and
predictable, with a focus on athletic fashions for everyday
people, much like how cars have become a ubiquitous part
of modern life.  The book should offer an entertaining and
engaging escape from reality, much like how a filmmaker can
transport the audience to another world through the magic of
cinema.
```

**The Corresponding Response:**

```
LLM Generates Knowledge is:   X_IK
– Look for a book that features an uplifting and
light-hearted storyline, where characters find joy in their
daily routines, creating a sense of comfort and familiarity.
– Seek narratives that emphasize the importance of community
and friendship, perhaps showcasing a diverse cast that
showcases various athletic fashions and lifestyles,
cheerfully blending sports with daily life.
– Consider selections that utilize humor and warmth
throughout the narrative, allowing readers to enjoy
predictable plot twists that evoke feelings of nostalgia and
contentment, similar to classic feel-good films.
– Explore themes revolving around self-acceptance and
empowerment in the context of athletic fashion, where
characters confidently navigate their own styles while
participating in community events or activities.
– Identify tales that incorporate scenic descriptions of
urban or rural settings, inviting readers to visualize
a vibrant world beyond their own, akin to the vivid
storytelling found in cinema.
– Make sure the story maintains a reassuring tone, providing
an optimistic perspective on life's challenges, akin to the
feel-good resolutions often found in popular films.
```

---

[16]https://www.perplexity.ai/

B.1.2 RERANKER

---

**The Prompt Template in Reranker:** $P_{tr}$

Based on the information, give recommendations for the user based on the constrains. Don't use numerical numbering for the generated content; you can use bullet points instead.
Candidate Ranking List $X_{Item}$: item id:96578, corresponding title:Surrender, Dorothy: A Novel, description:["Elle Devastatingly on target.The New York Times ;item id:10837, corresponding title:The Block (Urban Books), description:[''] ;item id:58215, corresponding title:Ritual: A Very Short Introduction (Very Short Intr, description:["Barry Stephenson is Assistant Professor of Relig ;item id:74947, corresponding title:The Collins Case (Heartfelt Cases) (Volume 1), description:['Julie C. Gilbert enjoys writing science fiction, ;item id:173346, corresponding title:Love Handles (A Romantic Comedy) (Oakland Hills), description:['Gretchen Galway is a USA TODAY bestselling autho ;item id:66448, corresponding title:Much Laughter, A Few Tears: Memoirs Of A WomanS Fr, description:[''] ;item id:174617, corresponding title:Drinking at the Movies, description:['', 'Lizzy Caplan Reviews Drinking at the Movies' ;item id:37955, corresponding title:Eternal Now (scm classics), description:["These 16 sermons contain in concentrated form so ;item id:59337, corresponding title:The Guy to Be Seen With, description:["Coming from two generations of journalists, writ ;item id:110713, corresponding title:A Merry Little Christmas: Songs of the Season, description:["Anita Higman is the award-winning author of more ,
Knowledge:Above Generated Knowledge, Static Interest $X_{SU}$:user historical information, item title:The Executive's Decision: The Keller Family Series,item description:. She is a member of Romance Writers of America and Colorado Romance Writers. Visit her website at www.bernadettemarie.com for news on upcoming releases, signings, appearances, and contests.', '', ''] ;user historical information, item title:Gumbeaux,item description: instructional design content for Fortune 100 companies. Her book, Gumbeaux, received top honors in the 2011 Readers Favorite fiction contest. She lives in San Diego county with her husband Michael.'] ;user historical information, item title:The Hummingbird Wizard (The Annie Szabo Mystery Series) (Volume 1),item description:['', ''] ;user historical information, item title:Artifacts (Faye Longchamp Mysteries, No. 1),item description:['', ''] ;user historical information, item title:3 Sleuths, 2 Dogs, 1 Murder: A Sleuth Sisters Mystery (The Sleuth Sisters) (Volume 2),item description:['Maggie Pill is a lot like Peg Herring, only much cooler and more interesting.'] ; ...(Pruning.),

---

**The corresponding response:**

Reranking Results $R^*$: {173346, 96578, 110713, 10837, 66448, 74947, 58215, 59337, 174617, 37955}
**Answer:** 173346
**Initial Ranking List** $R$: [ 96578 10837 58215 74947 173346 66448 174617 37955 59337 110713]

### B.1.3 SELF-REFLECTION

Rerank list is out of the order, you should rerank the item from the pure ranking list. The previous list:. Therefore, try it again according the following information.

---

**Extra Prompt $P_{sr}$:**
Rerank list is out of the order, you should rerank the item from the pure ranking list. The previous list: Pure Ranking List. Therefore, try it again according the following information.
The rest content is same as the content in the reranker.

---

## B.2 Prompt Templates and Examples Response in i²Agent

### B.2.1 Generator

---

**Step One**

**The Prompt Template in Generator $P_{pr1}$:**

Here is the background of one user.
Static Interest
Please recommend one item for her.  The first one
title:  Sidewinder (The Halcyon Files Book 1) eBook,
descrition:[Nan].  The second one title:Sound Design:  The
Expressive Power of Music, Voice and Sound Effects in Cinema,
description:[Nan].

**The Corresponding Response:**

Recommended Item $X_G^T$ Based on the provided options and
considering a general interest in storytelling and cinematic
experiences, I recommend "Sound Design:  The Expressive Power
of Music, Voice and Sound Effects in Cinema".
This book delves into the role of sound in film, exploring
how music and sound effects enhance narrative and emotional
depth.  It could provide valuable insights into the art of
storytelling through sound, making it a captivating read for
anyone interested in filmmaking or audio artistry.

**Step Two**

**The Prompt Template in Generator $P_{pr2}$:**

Great!  Actually, this user choose the item with title:
Sidewinder (The Halcyon Files Book 1) eBook and give the
review:  lots of twists and turns.  Can you generate the
profile of this user background?  Please make a detailed
profile.  Don't use numerical numbering for the generated
content; you can use bullet points instead.  .

**The Corresponding Response:**

Generate Profile is $\mathcal{F}^T$: - **Name:** [Not Provided]
- **Reading Interests:** Enjoys engaging and complex
narratives, particularly those with unexpected twists and
turns.
- **Preferred Genres:** Likely enjoys thriller, mystery, and
possibly science fiction or fantasy, given the title of the
chosen book, "Sidewinder (The Halcyon Files Book 1)".
- **Favorite Themes:** Thrill-seeking adventures, intricate
plots, and character development that surprises or challenges
expectations.
- **Typical Reading Habits:** Prefers books that keep them
guessing, possibly reading at night or during quiet moments
to fully immerse in the story.
- **Current Literary Explorations:** Currently venturing into
series that promise suspense and depth, likely seeking to
uncover hidden layers within the narrative.
- **Community Engagement:** Might engage with online
book clubs or forums to discuss their favorite twists in
literature and share recommendations.

---

### B.2.2 EXTRACTOR

**The Prompt Template in Generator:**
Based on the generated knowledge and the instruction, extract
some dynamic interest information from the static memory.
Moreover, based on the profile and the instruction, extract
some dynamic profile information.
Don't use numerical numbering for the generated content; you
can use bullet points instead.
Generated Knowledge:Knowledge,
Instruction:Instruction
Static Memory:Static Interest
Profile:Generated Profile .
**The Corresponding Response:**
Dynamic Interest $X_{DU}$:
- Look for uplifting narrations that are light-hearted and
similar to feel-good films, featuring characters finding joy
in simple outcomes.
- Seek books focused on themes of community and friendship,
where relationships flourish in relatable contexts, akin to
feel-good cinematic experiences.
- Explore narratives set in vibrant, adventurous locations,
enhancing the escapist feel of the story, reminiscent of
compelling cinematography.
- Consider stories that integrate elements of athletic
fashion, reflecting everyday expressions parallel to
vehicles' cultural significance in modern life.
- Ensure a predictable storyline for easy immersion,
providing comfort much like traditional movie endings.
- Look for a blend of humor and warmth that echoes the spirit
of light-hearted comedies, making them enjoyable reads.
- Investigate books featuring authentic characters who
embrace their imperfections, promoting positive self-image
like many cinematic heroes.
Dynamic Profile $\mathcal{F}_d^T$:
- **Reading Interests:** Enjoys feel-good narratives with a
focus on community and humor.
- **Preferred Genres:** Likely leans towards contemporary
fiction, lighthearted romance, and narratives that offer
escape and positivity.
- **Favorite Themes:** Appreciates uplifting tales that
emphasize personal growth, camaraderie, and authenticity in
character portrayals.
- **Typical Reading Habits:** Prefers reading during
relaxation times, seeking stories that provide a comforting
escape from daily life, similar to watching feel-good films.
- **Current Literary Explorations:** Interested in finding
relaxed reads that blend fun and heartwarming elements,
potentially exploring themes surrounding fashion and personal
identity in a light context.
- **Community Engagement:** May participate in discussions
around light literature and feel-good storytelling, possibly
through book clubs or social media platforms.

## B.3 EXAMPLES OF DATASET

### B.3.1 EXAMPLES IN CONSTRUCTING DATASET

**The Prompt Template in Constructing Dataset:**
Given the user's review of an item, please mimic the user's instruction which accurately describes their needs.
When crafting each instruction, please make a conscious effort to incorporate a distinct action word or descriptive term that diverges from those showcased in the provided examples.
The reply content should follow the structure: Review text: Persona: Final Instruction: . You should give the initial instruction first based on the reviews and then polish the instruction via mocking the provided persona. But do not reveal the persona directly, just mock their potential writing style. Please provide the instruction based on the review text and decide whether the generated instruction can be used in the examples.
Here are some examples..
Don't use numerical numbering for the generated content; you can use bullet points instead.
1st Reviews Example: Keith Green was a pioneer in the field of Christian rock, and I have loved every album he did. This one is particularly sweet as he was just coming into his own as a premier music writer and performer when it was published. His loss was a terrible blow for millions of his fans.
1st Personas Example: A music industry professional with a keen interest in developing new platforms for learning.
1st Instruction Example: I'm looking for an exceptional Christian rock album by Keith Green, especially one that showcases his emergence as a premier music writer and performer. His music has a special place in my heart, and something from his prime would be ideal.
2nd Reviews Example: I enjoyed the portraits of the heroine going through different transformations: the village girl to the servant to the prostitute to the library clerk...The novel seemed like a picaresque novel from the point of view of an Indian woman: sort of a mash-up of The Little Princess with Vanity Fair. The Pom to Sara to Pamela to Kamala roller coaster starts to become unbelievable towards the end, as the author doesn't spend as much time with the hero's transformation from colonialist to open-hearted husband.
2nd Personas Example:A data-driven finance officer responsible for allocating the school district's annual budget.
2nd Instruction Example: Seeking a novel that vividly portrays a heroine's transformative journey through various roles, akin to a picaresque tale from an Indian woman's perspective, blending elements of The Little Princess and Vanity Fair. Preferably, the narrative should effectively balance the heroine's evolution with the hero's significant transformation, exploring themes of power dynamics and their impact on relationships.
Other few-shot examples.
The User's Review:

### B.3.2 EXAMPLES OF FILTERED INSTRUCTIONS

We use an LLM to filter out instructions that may lead to data leakage. The following examples illustrate some of the filtered instructions.

---

**Some Filtered Instructions Examples:**

1st example:  As a ticket vendor, I am always on the lookout for a fascinating read that can provide a break from the routine, much like how I seek out the latest comedy films for a good laugh.  A book that offers a detailed look into WW2 submarine construction is what I crave.  However, I seek a book with clear and detailed photos and drawings, allowing me to fully appreciate the subject matter.  The book should be as captivating as a great comedy, providing a mix of entertainment and insight.  And just like how I appreciate a good joke, I seek a book that offers a satisfying read, leaving me feeling entertained and informed.  The book should leave me feeling like I have learned something new, much like how a successful comedy film can leave a ticket vendor feeling accomplished and motivated to recommend it to others.

2nd example:  In search of a book that offers a comprehensive and insightful look at the genre of mystery novels, much like how a dedicated science blogger can appreciate the intricacies of conducting precise experiments, I seek a narrative that captures the essence of the genre.  The book should offer a fresh perspective on the history and evolution of mystery novels, providing a realistic and engaging portrayal of the genre's development.  The narrative should be well-written and immersive, offering a depth and complexity that rivals the intricacies of conducting scientific experiments.  The book should also offer a nuanced exploration of the challenges and rewards of writing mystery novels, much like how a science blogger can delve into the intricacies of their field of study.

3rd example:  In my search for a book that can offer a fresh and insightful perspective on personality types and relationships, much like how a college professor recovering from a major accident can appreciate the value of alternative medicine, I seek a narrative that can challenge my assumptions and broaden my horizons.  The book should offer a well-researched and thoughtful analysis of personality types, much like how a college professor can appreciate the value of evidence-based research.  The author should also provide a sense of connection and understanding, much like how a college professor can find value in the human experience and the importance of relationships.  A book that meets these criteria would be a valuable addition to any reader's collection, offering a rich and rewarding reading experience that can inspire and inform.

---

### B.3.3 EXAMPLES OF RETAINED INSTRUCTIONS

The following examples show the retained instructions.

---

**Some Retained Instructions Examples:**

1st example: In my search for a book that offers a well-researched and informative narrative, much like how a child development researcher can appreciate the nuances of a well-written story that offers accurate and evidence-based information, I seek a resource that offers a comprehensive and engaging look at the subject matter. The book should feature a well-crafted plot that offers a rich history and background, much like how a child development researcher can appreciate the intricacies of a well-written story that offers accurate and evidence-based information. In short, I am seeking a book that offers a comprehensive and informative reading experience, much like how a child development researcher can appreciate the nuances of a well-written story that offers accurate and evidence-based information.

2nd example: In my search for a book that offers a source of motivation and inspiration, much like how a fellow naval officer with a strong background in logistics and supply chain management collaborates with a young officer on various projects to achieve success, I seek a narrative that can provide a compelling reading experience. The book should be a well-worn companion, offering insights and strategies for building and maintaining a successful career. The writing should be clear and concise, offering a reading experience that is as supportive as a mentor's guidance. And the narrative should offer a balance of action and introspection, much like how a naval officer seeks to balance the practical aspects of their work with a deeper understanding of the complexities and challenges of achieving success. The overall experience should be informative and thought-provoking, much like how a naval officer seeks to gain a deeper understanding of the challenges and opportunities of their career.

3rd example: In my pursuit of a book that offers a comprehensive guide to business continuity strategies, much like how a strategic planner approaches their work with precision and attention to detail, I seek a narrative that covers all aspects of planning and implementation. The book should be a source of guidance for those who seek to protect their organization from unexpected disruptions, offering a detailed examination of the latest techniques and approaches for ensuring business continuity. A book that meets these criteria would be a valuable addition to my collection, offering a thought-provoking and engaging read that can be enjoyed again and again. However, I request that the list provided to me be accurate and up-to-date, and that any books received in error be returned promptly and without hassle.

---

