# OpenReview forum: "iAgent: LLM Agent as a Shield between User and Recommender Systems"
_ICLR.cc/2025/Conference — Submitted to ICLR 2025_

### Official Review · Reviewer_reNA · 2024-10-17

**Soundness:** 2
**Presentation:** 3
**Contribution:** 2
**Rating:** 3
**Confidence:** 5

**Summary:**

This work aims to address the challenges of users objectively accepting the recommendation result from commercial platforms and propose an LLM-empowered agent standing between the user and the platform, acting like a shield. This work first proposes a simple iAgent as an initial solution, followed by a more sophisticated i²Agent. Further, the author proposes a dataset InstructRec and conduct experiments on it.

**Strengths:**

1. The motivation of LLM stands between the user and platform in recommender system is good.
2. The paper is well-written with a clear methodology.

**Weaknesses:**

1. iAgent shares many similarities with existing agents in recommendation, such as RecMind, RAH, InteRecAgent, and AgentCF. However, the authors do not clearly articulate the significant differences beyond the task setting. Since existing agent frameworks can easily adapt to different tasks by changing prompts, the lack of discussion on this aspect raises questions about the contribution of this work.
2. To my knowledge, iAgent appears to simply adapt existing architectures (e.g., InteRecAgent and RecMind). It consists of two main components: LLM-based user query analysis and prompt-based reranking (with self-evaluation essentially functioning as majority voting-based reranking). This design paradigm for agents is already common in the context of LLM as in-context recommenders (e.g., LLMRec, etc.) and LLM-based Agents (e.g. InteRecAgent and RecMind). Moreover, if i²Agent consistently outperforms iAgent, it suggests that iAgent should be part of i²Agent rather than having its own section.
3. For i²Agent, the core component is the Profile Generator, which is not particularly innovative. Using ground truth items to adjust an LLM’s personalized preference cache has already been applied in previous works, such as RAH and AgentCF, as early as last year. This raises doubts about the originality of the work.
4. In the experimental section, iAgent lacks comparisons with existing similar approaches, such as RecMind, RAH, and InteRecAgent, making its effectiveness questionable.

**Questions:**

please refer to weaknesses

---

### Official Review · Reviewer_kjVV · 2024-10-29

**Soundness:** 2
**Presentation:** 1
**Contribution:** 2
**Rating:** 3
**Confidence:** 4

**Summary:**

This paper proposes a novel idea: agent as the shield between recommender systems and users. While this idea is interesting, the proposed components do not address the novelty of the proposed idea. Meanwhile, this paper lacks discussion with related works and the presentation of this paper is poor.

**Strengths:**

1. The proposed agent-as-shield idea is interesting.
2. The paper is easy to understand.
3. The experiments in this paper are comprehensive.

**Weaknesses:**

1. The technical novelty of this paper is limited. Almost all the components (i.e., parser, reranker, memory, and self-reflection) in this paper are existing technologies [1, 2, 3]. The main novelty of this paper lies in the proposed concept of agent-as-shield, while this novelty is not reflected in the component design. The model design is not relevant to the agent-as-shield idea. It looks like a general agent framework, rather than agent-as-shield framework.

2. This paper lacks sufficient discussion and comparison with previous related works, and chooses troublesome baselines (i.e., instruction-aware methods). Therefore, the experiments are unable to support the claims of this paper.

    2.1 While using agents as a shield between the recommender system and users is an interesting research direction, this idea is similar to the RecSys-Assistant-Human framework proposed in RAH [4]. However, the paper does not address it.

    2.2 Moreover, this paper chooses troublesome baselines such as BM25, BGE-Rerank, and EasyRec, which have no relevance to this agent-based work, while ignoring agent-based baselines such as InteRecAgent. Therefore, the effectiveness of the proposed method remains unverified.
3. The writing of this paper can be significantly improved.

    3.1. The notations in this paper are chaotic. For example, all the notations in this paper adopt the same format and font, and the meaning of each notation is hard to understand (e.g., X_{SU}, P_{pr1}, and P_{pr2}). Please refer to prior works [5, 6] for guidance on presenting formulas more clearly and formally.

    3.2. It is unclear whether iAgent and i^{2}Agent are proposed separately. iAgent is a simplified and naive version of i^{2}Agent, and its purpose and meaning are not adequately explained.

    3.3. The task definition is unclear. First, the task definition from lines 143 to 149 is too short and ambiguous, which makes it hard to showcase the novelty of this task. Second, using “our task” to denote the proposed agent-as-shield task is ambiguous. Please use a specific task name for clarification. Third, defining RecAgent as a task name may lead to confusion, since it is also a method [7].

[1] Park, Joon Sung, et al. "Generative agents: Interactive simulacra of human behavior." *Proceedings of the 36th annual acm symposium on user interface software and technology*. 2023.

[2] Hou, Yupeng, et al. "Large language models are zero-shot rankers for recommender systems." *European Conference on Information Retrieval*. Cham: Springer Nature Switzerland, 2024.

[3] Zhao, Yuyue, et al. "Let me do it for you: Towards llm empowered recommendation via tool learning." *Proceedings of the 47th International ACM SIGIR Conference on Research and Development in Information Retrieval*. 2024.

[4] Shu, Yubo, et al. "RAH! RecSys–Assistant–Human: A Human-Centered Recommendation Framework With LLM Agents." *IEEE Transactions on Computational Social Systems* (2024).

[5] He, Xiangnan, et al. "Lightgcn: Simplifying and powering graph convolution network for recommendation." *Proceedings of the 43rd International ACM SIGIR conference on research and development in Information Retrieval*. 2020.

[6] Qian, Chen, et al. "Chatdev: Communicative agents for software development." *Proceedings of the 62nd Annual Meeting of the Association for Computational Linguistics (Volume 1: Long Papers)*. 2024.

[7] Wang, Lei, et al. "Recagent: A novel simulation paradigm for recommender systems." *arXiv preprint arXiv:2306.02552* (2023).

**Questions:**

Why do we need both iAgent and i^{2}Agent? I think iAgent is just a simplified version of i^{2}Agent, which is unnecassary for introducing.

I am happy to increase the score if the author can address my questions.

---

### Official Review · Reviewer_yPLW · 2024-10-31

**Soundness:** 2
**Presentation:** 3
**Contribution:** 2
**Rating:** 5
**Confidence:** 3

**Summary:**

This paper introduces a new user-agent-platform paradigm for recommender systems, where an agent acts as a shield between the user and the platform's algorithms. The authors construct INSTRUCTREC datasets that include user instructions along with interactions, and propose an Instruction-aware Agent (iAgent) that leverages these instructions to refine recommendations. They further develop an Individual Instruction-aware Agent (i2Agent) that optimizes each user's unique profile and interests using a dynamic memory mechanism. Experiments show the effectiveness of the proposed method.

**Strengths:**

* The authors' development of the INSTRUCTREC datasets, which incorporate user instructions alongside user-item interactions, provides a valuable new benchmark for evaluating recommendation agents.
* The paper introduces two agent-based models, iAgent and i2Agent, that demonstrate strong performance improvements over state-of-the-art baselines.

**Weaknesses:**

* The generated dataset is relatively small, with each subset containing fewer than 100,000 entries. This may limit the robustness of the findings. As a result, traditional recommendation models may be trained in an underfit manner. In Table 4, some performance of traditional recommendation models resembles random guessing.


* The paper does not compare its approach with existing LLM-based recommendation methods, such as TALLRec [1] and LLaRA [2].

* It is recommended to report token consumption and model complexity for this work, as these factors are essential for understanding the computational demands of the proposed models.


* The agent-based approach may introduce new types of biases, such as the agent's own inherent biases based on its training data and knowledge sources. Evaluating and mitigating such biases should be considered.


Reference
[1]Bao, K., Zhang, J., Zhang, Y., Wang, W., Feng, F., & He, X. (2023, September). Tallrec: An effective and efficient tuning framework to align large language model with recommendation. In Proceedings of the 17th ACM Conference on Recommender Systems (pp. 1007-1014).
[2]Liao, J., Li, S., Yang, Z., Wu, J., Yuan, Y., Wang, X., & He, X. (2024, July). Llara: Large language-recommendation assistant. In Proceedings of the 47th International ACM SIGIR Conference on Research and Development in Information Retrieval (pp. 1785-1795).

**Questions:**

Please refer to the "Weaknesses".

---

### Official Review · Reviewer_pJpw · 2024-11-02

**Soundness:** 2
**Presentation:** 3
**Contribution:** 2
**Rating:** 5
**Confidence:** 4

**Summary:**

This paper presents a novel method called iAgent that applies LLMs to recommender systems. This approach adopts a user-centered perspective by creating a unique agent for each user as the recommendation model. Specifically, the authors convert users' historical actions into instructions and employ multiple LLMs, such as a Parser, Extractor, and ReRanker, to model user behavior. This approach captures user preferences from the user’s perspective rather than the platform’s. The authors also provide a recommendation dataset containing user instructions and conduct extensive experiments to evaluate the method's effectiveness.

**Strengths:**

1. The paper is clearly written and easy to understand. The appendix includes extensive experimental details, allowing future researchers to easily follow and replicate the study.
2. Adopting LLMs for re-ranking is quite novel, as they can leverage world knowledge to enhance result interpretability and improve re-ranking accuracy.

**Weaknesses:**

1. The LLMs used in this paper are zero-shot without fine-tuning, which relies heavily prompt engineering. Additionally, all four datasets are based on user reviews. In other domain-specific scenarios where world knowledge is less relevant, these zero-shot LLMs may perform poorly.
2. Personalized re-ranking is quite common in modern recommender systems. For example, [1-3] proposed re-ranking models that adjust the initial ranking list based on user preferences. The authors should compare their LLM-based re-ranking method with these non-LLM-based approaches.
3. The authors randomly sample 9 negative items along with one true item to create the candidate ranking list (L320). This distribution differs significantly from the actual ranking output of a real ranking model. Some learning-to-rank (LTR) models could be used instead to generate a more realistic initial ranking list.
4. Compared to other LLM-based methods, it’s unclear where the primary performance gains of this approach are from. For example, are the gains due to the Profile Generator’s ability to capture user traits or the Parser’s richer information extraction? How do different prompts affect metrics like HR? A more detailed ablation study should be added.
5. The authors mention the decline in LLM performance with longer sequence length (L422) but do not adequately solve it. Additionally, they select users and items with sufficient behavioral data (L293), which may further intensify this issue.
[1] Personalized Re-ranking for Recommendation
[2] Multi-factor Sequential Re-ranking with Perception-Aware Diversification
[3] On-device Integrated Re-ranking with Heterogeneous Behavior Modeling

**Questions:**

1. Have the authors tested on long-tail users and long-tail items (e.g., with fewer than 5 actions)? Since LLMs leverage world knowledge, they might be particularly useful for cold-start scenarios.
2. Could the random assignment of user personas cause mismatches between user profiles and their historical behaviors? (L299)

---

### Author Response · Authors · 2024-11-23
**General Response by Authors**

Dear Area Chairs and Reviewers,

We appreciate the reviewers' time, valuable comments, and constructive suggestions. The reviewers recognized our critical contributions in developing novel agent-based recommendation models (yPLW) and introducing the innovative agent-as-shield approach (kjVV). They highlighted the value of our INSTRUCTREC datasets as a new benchmark (yPLW) and our effective use of LLMs for re-ranking to enhance result interpretability (pJpw). The reviewers found our paper well-written (pJpw, reNA) with comprehensive experiments (kjVV) and clear methodology (reNA).

We provide the response to the reviewer's general problems.

> ***"Q1.Why don't our baselines include more recommendation agent methods?"***

In our paper, we review five recommendation agents [1-5] in the related work section and select two of them for our experimental evaluation. We did not compare against all methods primarily because some authors did not release their source code, and their experimental datasets (100-1,000 samples) represent only approximately 3% of our InstructRec dataset (~30,000 samples). Using such small datasets would lead to prohibitively expensive API costs for large language models, as scaling up to our dataset size would increase costs by roughly 30 times.

In future work, we plan to conduct more comprehensive experiments incorporating additional baseline methods, utilizing open-source models that can be run locally to avoid API costs.

> ***"Q2. What distinguishes iAgent's novelty from existing recommendation agents?"***

First, as acknowledged by reviewers yPLW and kjVV, we introduce a novel perspective on user protection in recommender systems, aiming to mitigate both the echo chamber effect and potential platform manipulation.

Second, reviewer reNA seems to have misunderstood the novelty of i$^2$Agent. Our profile generator creates personalized memory for each user. Another crucial component of i²Agent is the dynamic extractor, which selectively retrieves the most relevant content from users' memory combined with enhanced internal and external knowledge. This dynamic extraction mechanism has been rarely explored in previous work.

Third, to promote research in this field, we contribute a benchmark dataset for instruction-aware recommendation agents and make our implementation publicly available.

[1] Huang, Xu, et al. "Recommender ai agent: Integrating large language models for interactive recommendations." arXiv preprint arXiv:2308.16505 (2023).

[2] Wang, Zhefan, et al. "Multi-Agent Collaboration Framework for Recommender Systems." arXiv preprint arXiv:2402.15235 (2024).

[3] Wang, Yancheng, et al. "Recmind: Large language model powered agent for recommendation." arXiv preprint arXiv:2308.14296 (2023).

[4] Zhang, An, et al. "On generative agents in recommendation." Proceedings of the 47th international ACM SIGIR conference on research and development in Information Retrieval. 2024.

[5] Zhang, Junjie, et al. "Agentcf: Collaborative learning with autonomous language agents for recommender systems." Proceedings of the ACM on Web Conference 2024. 2024.

---

### Meta-Review · Area_Chair_p8Fh · 2024-12-21

**Metareview:**

This paper proposes constructing an LLM agent to act as a shield between the user and recommender system to address challenges like echo chamber effects in recommendation scenarios. The paper's motivation was recognized by most reviewers, and this paper is clear and easy to understand. However, reviewers also pointed out some weaknesses, and the most important two are: 1) The method lacks novelty, as most of the components have already been used in existing studies; 2) The choice of baselines for experimental comparisons is lacking, and the experiments are insufficient to fully validate the method's effectiveness. All reviewers think this paper is not ready by far, so our decision leans toward rejection.

**Additional Comments On Reviewer Discussion:**

During the rebuttal phase, the authors did not specifically address each concern raised by the reviewers. They only provided some responses to the two major challenges, but these responses did not convince the reviewers or me.

---

### Decision · Program_Chairs · 2025-01-22

Reject